# Identification of the Third Case of PSEN1 Tyr389His Variant in Early-Onset Alzheimer’s Disease in Korea

**DOI:** 10.3390/ijms232416192

**Published:** 2022-12-19

**Authors:** Kyu Hwan Shim, Sangjoon Kang, Seong Soo A. An, Min Ju Kang

**Affiliations:** 1Department of Bionano Technology, Gachon University, Seongnam-si 13120, Republic of Korea; 2Department of Neurology, Veterans Medical Research Institute, Veterans Health Service Medical Center, Seoul 05368, Republic of Korea

**Keywords:** Alzheimer’s disease, PSEN1, Tyr389His, exome sequencing

## Abstract

Amyloid precursor protein (*APP*), presenilin 1 (*PSEN1*), and presenilin 2 (*PSEN2*) are associated with autosomal-dominant early-onset Alzheimer’s disease (AD). Most mutations have been identified in the *PSEN1* gene. We discovered a *PSEN1* mutation (Tyr389His) in a Korean patient with early-onset AD who presented memory decline at 41 years of age followed by language, memory, and visuospatial dysfunctions. As this is the third such patient identified in Korea, this mutation may be involved in AD pathogenesis, suggesting that routine screening is necessary in this population. Altered intra-molecular interactions with the mutated amino acid may result in the destabilization of γ-secretase. In the future, a panel incorporating genes with relatively high-frequency rare variants, along with the *APOE4* gene, may predict the onset of AD and facilitate customized treatment.

## 1. Introduction

Accumulation of misfolded amyloid beta (Aβ) and tau proteins in amyloid plaques and neuronal tangles are pathological characteristics of Alzheimer’s disease (AD). In the amyloidogenic pathway, amyloid precursor protein (APP) is cleaved by β-secretase and γ-secretase complexes to form Aβ peptides [1]. Presenilin-1 gene (*PSEN1*) mutations are the most often reported in patients with early-onset AD (EOAD). Over 300 *PSEN1* mutations in the γ-secretase complex have been reported as regulating APP processing through their effects on the increased formation of insoluble toxic Aβ fibrils in plaques [2]. *PSEN1* mutations might change the structure of γ-secretase by inhibiting the first endoproteolysis (ε-cleavage) step and increase the release of intermediate products, such as longer amyloid peptides [3]. *PSEN1* mutation-associated EOAD typically manifests between the ages of 40 and 60. The world’s fastest-aging countries, namely China, Korea, and Japan, are bracing to have a dramatic increase in AD patients. According to a literature analysis, more than 55 *PSEN1* mutations were discovered in Asia [4]. The highest number of mutations (15 in total) were found in exon 7 of *PSEN1*, encompassing the transmembrane (TM) domains-IV, V, and VI. A substantial majority of pathogenic mutations have been identified in the TM domain [5] and catalytic aspartate residues (Asp257, 384) are located in TM-VI and VII [6].

Herein, we report the clinical phenotype of a patient with the *PSEN1* Tyr389His variant. Interestingly, this rare variant was recently reported in two EOAD patients in Korea [7,8]. Hence, this report reinforced the significance and pathogenicity of the *PSEN1* Tyr389His variant in Korea through in silico analysis for the prediction of structural changes.

## 2. Results

### 2.1. Clinical Characterization

A 46-year-old right-handed, high school-graduated female presented with a 5-year history of progressive short-term memory decline. She had difficulty recalling names and the way home. However, the patient could manage her daily activities. Her family history showed that her mother exhibited cognitive impairment in her 30s and died in her 40s. Her father remained neurologically normal until his death in his 80s. All three of her siblings were healthy and cognitively normal. All living family members refused to undergo genetic testing. She scored 21/30 on the Mini-Mental State Examination. Her global clinical dementia rating was 0.5. The sum of the box scores was 1.5. Neuropsychological testing revealed dysfunctions in language, memory, and visuospatial abilities. MRI revealed minimal cortical atrophy (Figure 1A) without further changes since the first visit. [^18^F] FDG-positron emission tomography (PET) revealed mild hypometabolism in the bilateral parietotemporal areas (Figure 1B). Furthermore, 18-Florbetaben PET also revealed positive findings, with a brain Aβ plaque load (BAPL) score of 3 (Figure 1C). Additional analysis for cognitive impairment due to APOE genetic factors revealed that this patient was of the ε3/ε3 type.

### 2.2. Genetic Analysis and In Silico Prediction

Upon whole-exome sequencing, tyrosine was found to be replaced with histidine at codon 389 (Tyr389His) in presenilin-1 protein. In addition, leucine was substituted with methionine at codon 1518 in exon 25 (Leu1518Met) of NOTCH3 protein (Figure 1D). Other rare variants found in neurodegeneration-related genes, such as *ALS2*, *GAB2*, and *SORL1*, might contribute to AD pathophysiology in association with *PSEN1* and/or *NOTCH3* (Figure 1F).

The prediction of structural alteration was conducted since *PSEN1* mutation could induce destabilization of γ-secretase through changes in intra-molecular interactions. Structure prediction supported the pathogenicity of the variant, as the absence of the hydrogen bonding of His389 affected the position changes against Leu435 and Ser438 amino acids (Figure 1E). Asp385, Phe386, Leu392, and Val393 were predicted to interact with both wild and mutant amino acids and showed minor structural changes in the mutant presenilin-1 protein.

## 3. Discussion

Previously identified *PSEN1* variants show strong pathogenicity in AD development and are also known to be the most prevalent pathogenic variants [9]. We detected a *PSEN1* variant, Tyr389His, in a Korean patient with EOAD. *PSEN1* Tyr389His has been previously reported in two Korean AD patients [7,8] with an age of onset below 40 years and a strong family history supporting the *PSEN1* Tyr389His variant’s pathogenicity. Since our patient already had symptoms for the past 5 years, the age of onset would be similar to that in the previous cases. To our knowledge, this is the third report from Korea detailing the *PSEN1* Tyr389His variant in an EOAD patient. All three cases with the *PSEN1* Tyr389His variant were reported in Korea only, and all patients had EOAD. This evidence supports the fact that the *PSEN1* Tyr389His variant has strong pathogenicity in AD development and higher frequency, especially in Korea.

Presenilin-1 protein has nine TM domains, which are known as membrane-associated hydrophobic domains. Most previously reported pathogenic *PSEN1* mutations were located between TM-II and TM-VII. *PSEN1* Tyr389His was found in the TM-VII region. The mutation site is evolutionarily conserved (GERP score = 4.54), and in silico prediction suggested it is most probably damaging (Polyphen2) and not tolerated or damaging (SIFT) [7,8]. Furthermore, it also has a CADD score of 26.9, indicating that it is in the top 1% of detrimental mutations. In our analysis, we predicted that this mutation might affect the destabilization of γ-secretase and Aβ production through structural changes in the TM-VII region of the protein (Figure 1E). The decreased hydrophobic interactions between Tyr and His could also influence the dynamics of the transmembrane helix, perhaps leading to increased Aβ production.

*NOTCH3* mutations are primarily responsible for the development of cerebral arteriopathy with subcortical infarcts and leukoencephalopathy (CADASIL). *NOTCH3* Leu1518Met was previously reported in a Korean patient with white-matter hyperintensities in the bilateral external capsule and temporal poles [10]. A recent study revealed that rare *NOTCH3* variants, including Leu1518Met, may contribute to elevating the white-matter hyperintensities in Parkinson’s disease [11]. Despite being a rare *NOTCH3* variant, our patient exhibited no CADASIL-related symptoms, stroke, or white-matter intensity changes. However, owing to the close interactions between *PSEN1* and *NOTCH3*, it would be difficult to rule out the possibility of *NOTCH3* involvement in disease progression (Figure 1F).

Various mutations associated with EOAD have been identified; however, their clinical implications are yet to be determined. Recently, attempts have been made to predict and prevent disease by a mutation panel related to AD, as well as sequencing the *APOE4* gene. The prediction of the onset of AD through the gene panel may allow the use of personalized medicine for early management of the disease based on the patient’s mutation status. To establish a precise gene panel, it is necessary to clarify the pathogenicity of the identified rare variants through various methods, such as functional studies of family history and the higher frequency in AD patients. It should also be considered that the frequency of each mutation varies by race or country. 

## 4. Method

### 4.1. Subject

The study participant gave written informed consent for the use of genetic and clinical data for research purposes. National Institute on Aging–Alzheimer’s Association (NIA-AA) criteria were used to determine a probable AD diagnosis. This study was approved by the Institutional Review Board of Veterans Healthcare Medical center (15 June 2021).

### 4.2. DNA Purification and Genetic Screening

Whole blood was drawn from the patient using an EDTA tube and kept at −20 °C until used. A QIAamp DNA Blood Maxi Kit (Qiagen, Hilden, Germany) was used to purify genomic DNA following the manufacturer’s instructions. To confirm the presence of the mutation, PCR was performed in both directions, followed by Sanger sequencing of the products (Bioneer Inc., Daejeon, Korea). An ABI 3730XL DNA Analyzer was used to process BigDye Terminator Cyclic sequencing. The gene and protein sequences were validated against the NCBI Gene (http://www.ncbi.nlm.nih.gov/gene (accessed on 13 July 2022)) and UniProt (http://www.uniprot.org (accessed on 13 July 2022)) databases.

### 4.3. In Silico Analyses

Structural changes in presenilin-1 protein caused by the mutation were examined to assess its potential pathogenicity Missence3D (http://missense3d.bc.ic.ac.uk/missense3d/ (accessed on 26 August 2022)) was used to estimate the 3D protein structure of normal and mutant presenilin-1 protein. Discovery Studio 3.5 Visualizer was used to process the visualization and interaction with the surrounding amino acids.

## 5. Conclusions

In the case of *PSEN1* Tyr389His, three patients with EOAD were reported in Korea only, a relatively higher frequency than that reported in other countries and races. Hence, this single-nucleotide polymorphism should be included in the Korean genetic screening panel for EOAD to enable the provision of appropriate genetic counseling and improve diagnostic accuracy. In the future, there is a need to study race- or country-specific mutations, which will help enable precise personalized medicine.

## Figures and Tables

**Figure 1 ijms-23-16192-f001:**
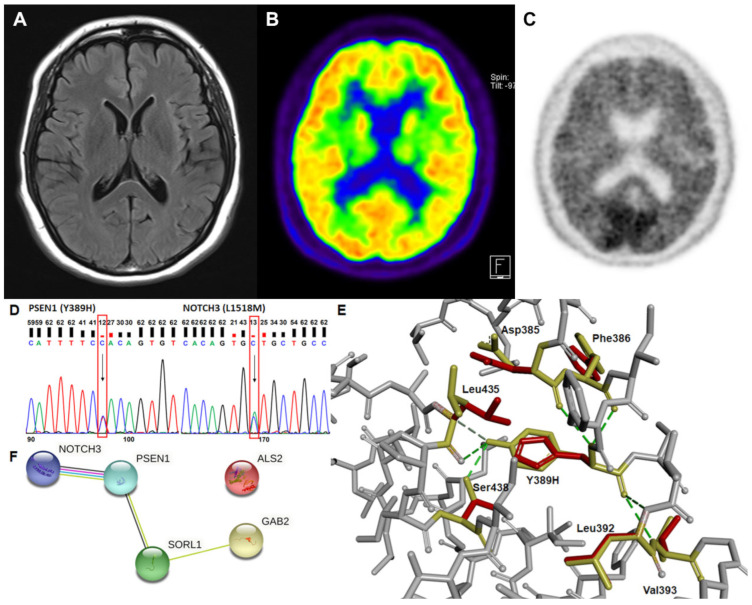
Brain images, prediction of gene interactions, and the alterations in the intermolecular interactions from the genetic analysis of a patient are presented. (**A**) The axial brain fluid-attenuated inversion recovery magnetic resonance image shows minimal cortical atrophy. (**B**) The axial 18F-fluorodeoxyglucose positron emission tomography image reveals mild hypometabolism in the bilateral parietotemporal areas. (**C**) The 18F-florbetaben positron emission tomography image shows positive findings. (**D**) Sanger sequencing of *PSEN1* and *NOTCH3* variant. Each heterozygous variant is indicated by a black arrow and a red box. (**E**) The structural changes due to the absence of hydrogen bonds of His389 with Leu435 and Ser438 amino acids in mutant presenilin-1 protein were predicted. The original amino acids and their positions are presented in yellow, while mutant His389 and the altered position of the related amino acids are in red. (**F**) Possible interaction among genes for which rare variants were found in a patient.

## Data Availability

The data presented in this study are available on request from the corresponding author.

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
