# Peer review of "Identification of the Third Case of PSEN1 Tyr389His Variant in Early-Onset Alzheimer’s Disease in Korea"

_ijms, 2022, doi:10.3390/ijms232416192_

Round 1

Reviewer 1 Report

In this manuscript, the authors describe the third Korean patient with early-onset Alzheimer disease (EOAD) associated with a PSEN1 Tyr389His mutation. The age of onset of this patient was 40 years of age, like the two previously described cases. Structural changes were determined by in silico analyses and are predicted to change intra-molecular interactions that result in destabilization of g-secretase. 

The findings presented in the manuscript reinforce the early assumption that this particular mutation causes EOAD, and in particular that the age of onset is predicted at around 40 years of age. 

The manuscript is well written, the results are well presented, and the discussion is thorough and pertinent. 

Major comment:  Figure 1 appears to be corrupted since it is missing its upper and bottom parts. This needs to be fixed prior to resubmission. 

Author Response

Response to Reviewer 1 Comments

In this manuscript, the authors describe the third Korean patient with early-onset Alzheimer disease (EOAD) associated with a PSEN1 Tyr389His mutation. The age of onset of this patient was 40 years of age, like the two previously described cases. Structural changes were determined by in silico analyses and are predicted to change intra-molecular interactions that result in destabilization of g-secretase.

The findings presented in the manuscript reinforce the early assumption that this particular mutation causes EOAD, and in particular that the age of onset is predicted at around 40 years of age.

Point 1: Major comment:  Figure 1 appears to be corrupted since it is missing its upper and bottom parts. This needs to be fixed prior to resubmission.

Response 1: Thanks for your correction. The figure was revised.

Reviewer 2 Report

The authors report the third case of a presenilin 1 mutation, which is found only in Korea. This is a strong indication of a likely pathogenic character of this mutation and it would be interesting to conduct a functional study on the protein to determine the effect of this mutation on protein function.

However, for the current manuscript, I have 3 comments:

1.    The introduction is too short and should include information relevant to the case report such as the previous 2 mutations found in Korea. A presentation of where most mutation of presenilin are located and their classification in pathogenic, likely pathogenic and unknown and the previous classification of this mutation is unknown according to the literature

2.    The most important information that is missing in the manuscript is the apoE genotype of the patient which is required to evaluate the likelihood that the changes in memory performance observed in the patient are not due to apoE genotype

3.    The authors report also the presence of an additional mutation in NOTCH3. It’s important to note that the same mutation Leu1518Met has been identified in a recent study involving Parkinson’s Disease patients (Ramirez et al., Movement Disorders 2020 PMID: 32573853). The authors need to include that information in the manuscript.

Author Response

Dear Editor and Reviewers,

We greatly appreciate your thoughtful comments that helped us to improve the manuscript's quality and readability. It is pleasure to receive positive feedbacks from all the reviewers. Upon reviewing our revisions, we hope that you will find the manuscript acceptable for publication in International Journal of Molecular Sciences.

Response to Reviewer 2 Comments

The authors report the third case of a presenilin 1 mutation, which is found only in Korea. This is a strong indication of a likely pathogenic character of this mutation and it would be interesting to conduct a functional study on the protein to determine the effect of this mutation on protein function.

However, for the current manuscript, I have 3 comments:

Point 1: The introduction is too short and should include information relevant to the case report such as the previous 2 mutations found in Korea. A presentation of where most mutation of presenilin are located and their classification in pathogenic, likely pathogenic and unknown and the previous classification of this mutation is unknown according to the literature

Response 1: Thanks for your kind opinion. We included more information in the introduction part. (p.#1, lines #32-#43)

Point 2: The most important information that is missing in the manuscript is the apoE genotype of the patient which is required to evaluate the likelihood that the changes in memory performance observed in the patient are not due to apoE genotype

Response 2: Thanks for raising an important point. The patient APOE type was ε3/ε3, and this was stated in the manuscript. (p.#2, lines #80-#81)

Point 3: The authors report also the presence of an additional mutation in NOTCH3. It’s important to note that the same mutation Leu1518Met has been identified in a recent study involving Parkinson’s Disease patients (Ramirez et al., Movement Disorders 2020 PMID: 32573853). The authors need to include that information in the manuscript.

Response 3: As reviewer’s suggestion, we added the recent study about Leu1518Met variant from PD in the discussion part. (p.#4, lines #133-#137)
